# Screening for Vaginal and Endocervical Infections in the First Trimester of Pregnancy? A Study That Ignites an Old Debate

**DOI:** 10.3390/pathogens10121610

**Published:** 2021-12-10

**Authors:** Leonie Toboso Silgo, Sara Cruz-Melguizo, María Luisa de la Cruz Conty, María Begoña Encinas Pardilla, María Muñoz Algarra, Yolanda Nieto Jiménez, Alexandra Arranz Friediger, Óscar Martínez-Pérez

**Affiliations:** 1Department of Obstetrics and Gynecology, University Hospital of Getafe, 28905 Madrid, Spain; 2Department of Obstetrics and Gynecology, Puerta de Hierro University Hospital of Majadahonda, 28222 Madrid, Spain; saracruz.gine@yahoo.es (S.C.-M.); beenpar@yahoo.es (M.B.E.P.); cushe2@gmail.com (Y.N.J.); alexandramigloo@yahoo.es (A.A.F.); oscarmartinezgine@gmail.com (Ó.M.-P.); 3Fundación de Investigación Biomédica, Puerta de Hierro University Hospital of Majadahonda, 28222 Madrid, Spain; farmcruz@gmail.com; 4Department of Microbiology, Puerta de Hierro University Hospital of Majadahonda, 28222 Madrid, Spain; algarra18@hotmail.com; 5Department of Obstetrics and Gynecology, Universidad Autónoma de Madrid, 28029 Madrid, Spain

**Keywords:** vaginosis, Ureaplasma, preterm birth, vaginal, infection, endocervical, screening, flora, pregnancy

## Abstract

Objectives: Vaginal and endocervical infections are considered a global health problem, especially after recent evidence of their association with preterm delivery and other adverse obstetric outcomes. Still, there is no consensus on the efficacy of a screening strategy for these infections in the first trimester of pregnancy. This study evaluated their prevalence and whether screening and treatment resulted as effective in reducing pregnancy and perinatal complications. Methods: A single-center prospective observational study was designed; a sample size of 400 first-trimester pregnant women was established and they were recruited between March 2016–October 2019 at the Puerta de Hierro University Hospital (Spain). They were screened for vaginal and endocervical infections and treated in case of abnormal flora. Pregnancy and delivery outcomes were compared between abnormal and normal flora groups by univariate analysis. Results: 109 patients had an abnormal flora result (27.2%). The most frequently detected infection was *Ureaplasma urealyticum* (12.3%), followed by *Candida* spp. (11.8%), bacterial vaginosis (5%), *Mycoplasma hominis* (1.2%) and *Trichomonas vaginalis* (0.8%). Patients with abnormal flora had a 5-fold increased risk of preterm premature rupture of membranes (5.3% vs. 1.1% of patients with normal flora, Odds Ratio 5.11, 95% Confidence Interval 1.20–21.71, *p* = 0.028). No significant differences were observed regarding preterm delivery or neonatal morbidity. Conclusions: Considering the morbimortality related to prematurity and that the results of our study suggest that the early treatment of abnormal flora could improve perinatal outcomes, the implementation of a screening program during the first trimester should be considered.

## 1. Introduction

Preterm birth is considered one of the main problems in modern obstetrics because is the second most frequent direct cause of death in children younger than 5 years old and it is the leading cause of perinatal morbidity and mortality in developed countries, in addition to causing astronomical economic costs. Despite medical advances and improved obstetric care, the rate of prematurity has been increasing during the last decades, being around 11–12% worldwide (with 15 million preterm children being born every year across the world), with great variability by geographical region (8–9% in developed countries) but also within countries according to income level, maternal education, race and other factors [1,2,3]. In 2005, the Institute of Medicine (IOM) estimated the cost of medical care for children born prematurely up to the age of 5 at $26 billion per year. This amount was underestimated because indirect costs were not included, such as the expenses that the families had to assume for hiring caregivers, transportation or home adaptations and the loss of income due to the fact that a parent in many cases was temporarily unavailable to return to work and the difficulty of employment in the case of premature children with disabilities [4].

Among the multiple factors described as causing prematurity, subclinical intraamniotic infection, produced by the ascent of germs from the lower genital tract is responsible for up to a third of premature births [5]. In 1995, the relationship between bacterial vaginosis (BV) and other vaginal and endocervical infections with the increased risk of preterm birth, was published for the first time [6]. In 2004, Kiss began to evaluate the suitability of vaginal and endocervical infection screening programs to reduce preterm birth [7], and in 2007, Leitich published a meta-analysis with more than 30,000 patients and established that bacterial vaginosis in asymptomatic patients significantly increases the risk of prematurity (OR 2.16, 95% CI 1.56–3.00), of late miscarriage (OR 6.32, 95% CI 3.65–10.94) and maternal infection (OR 2.53, 95% CI 1.26–5.08) [8]. Additionally, there is also evidence that intermediate flora (Nugent score between 4 and 6) may increase the risk of prematurity [9,10], and other endocervical germs have been associated with preterm birth, such as *Ureaplasma urealyticum* (UU) and *Mycoplasma hominis* (MH) [11,12,13]. Even recurrent candidosis, which is often neglected and untreated in the absence of symptoms, has been associated with an increased risk of prematurity in some studies [14].

Although there is data that suggest that early treatment of these vaginal infections seems to improve perinatal prognosis [15], there is contradictory evidence regarding the usefulness of antibiotic treatment [16,17,18,19], probably due to the heterogeneity of the studies and their methodological variability [20]. For these reasons, we consider necessary to study the prevalence of vaginal/endocervical infections in our pregnant women population and compare it with other similar studies that have shown a decrease in neonatal morbidity and mortality with antibiotic treatment. The assessment and evaluation of these findings in the Spanish population could lead to a substantial change in healthcare protocols.

## 2. Materials and Methods

### 2.1. Study Design and Participants

Our study was designed as a prospective observational single-center study. The study was approved by the Ethics Committee of Hospital Universitario Puerta de Hierro (November 2015). Between March 2016 and October 2019, obstetricians at the Puerta de Hierro University Hospital (Spain) provided first-trimester pregnant women with information about our study and offered them to participate.

The inclusion criteria were: women over the age of 18, with single pregnancies between 8 + 0 and 15 + 6 weeks, who agreed to receive the recommended treatment in case of an altered smear result, and who planned to complete the follow up of their pregnancy at our center (or in case of delivery in another center, agreed to be contacted by telephone to complete the data collection). The exclusion criteria were the inability to give informed consent, refusal to take smear samples or to take the antibiotic treatment indicated by the investigators, having been treated vaginally with progesterone in the last 7 days or having been treated with oral or vaginal antibiotics or antiseptics during the last month.

According to vaginal infections’ prevalence, values reported in the scientific literature [12,21,22], and using an expected error of 3–5% and a confidence level of 95%, a sample size of 400 women was established for the study. This estimation was carried out with the !Nsize macro run in SPSS V.20 (IBM Inc., Chicago, IL, USA).

### 2.2. Procedures

In case of having agreed to participate in the study, and once it was verified that they met the inclusion criteria and none of the exclusion criteria, women would sign the informed consent form. Then, one of the researchers proceeded to question the patients about their obstetric history and the presence of vulvovaginal symptoms (leucorrhoea, bad odor, or itching) and a gynecological examination was performed using a speculum to do a visual inspection and evaluate the presence of signs such as leucorrhoea, bad odor or erythema.

After the examination, two samples of vaginal smear (preferably from the area with the most discharge or, alternatively, from the posterior fornix), and two samples of endocervical smear were collected, with the purpose of screening the most common vaginal/endocervical infections.

The samples were sent to the Microbiology deparment where they were processed and reviewed by the same microbiologist in all cases. First, a fresh examination was performed for the visualization of yeasts, clue cells, leukocytes and Trichomonas. Then, a Gram stain and an interpretation of the Nugent criteria were performed for the diagnosis of bacterial vaginosis (determining the relative quantity of morphotypes characteristic of abnormal vaginal microbiota: Gram-positive and Gram-negative bacilli and curved bacteria). Finally, all the samples were cultured in the usual media conditions of the temperature and time of incubation recommended by the Spanish Society of Infectious Diseases and Clinical Microbiology (SEIMC): Columbia agar with 5% sheep blood, chocolate agar, Granada medium (for the isolation of *Group B streptococcus*), Chromogenic Candida agar, Thayer Martin medium (for the isolation of *Neisseria gonorrhoeae*), and Roiron medium (for the isolation of *Trichomonas vaginalis*). The incubation time until results obtained was 48 h in the case of vaginal smear and 72 h in the case of endocervical smear. For the *Mycoplasma* study, a *Mycoplasma* IST 2 kit (Biomerieux^®^) was inoculated, performing the reading at 24 and 48 h of incubation and informing the result as negative, presence (<10,000 UCC/mL) or positive (>10,000 UCC/mL) [23,24,25].

After obtaining the results of all the tests described above, the smears were classified into three categories, in order to relate them to eventual obstetric adverse events:-Normal or usual flora: Nugent score between 0 and 3 and no other pathogen isolated in the cultures;-Intermediate flora: Nugent score between 4 and 6 and no other pathogen isolated in the cultures;-Abnormal flora: Nugent score between 7 and 10 (and therefore diagnosis of bacterial vaginosis), or if another pathogen was isolated in the cultures.

In our study, abnormal flora was always considered as infection, regardless of the existence or not of associated signs or symptoms and adequate antibiotic treatment was prescribed based on the current clinical practice guidelines for pregnant women [26]. These are as follows:
○Bacterial vaginosis: 2% vaginal clindamycin (5 g) every 24 h for 6 nights;○*Ureaplasma/Mycoplasma* *: Azithromycin 1 g oral weekly for 2 weeks;○*Candida* spp.: Clotrimazole vaginal ovules 100 mg every 24 h for 6 nights;○*Neisseria gonorrhoeae* *: a single dose of Ceftriaxone 250 mg intramuscular;○*Trichomonas vaginalis* *: vaginal metronidazole 500 mg for 7 nights.


* In these cases, specific treatment to sexual partners was also prescribed to avoid recurrences.

After confirming the correct compliance of the treatment, the cultures were only repeated if symptoms appeared, as in the rest of the pregnant women according to standard clinical practice. No other procedures or face-to-face visits with the study patients were performed outside of the usual pregnancy follow up. 

### 2.3. Outcomes

In order to record the information needed, a specific database was designed for the study. The data was entered by the lead researcher after the delivery of each patient. The characteristics and obstetric history of the study patients were extracted from their clinical history. Alongside the vaginal and endocervical smears results, the following variables were recorded:
Gestational age, signs or symptoms at the time of sample collection (leukorrhea, erythema, bad odor, itching), vaginal discomfort with sample collection (referred by the patient from 1 to 5);Antecedents of interest: maternal age, active smoking habit (and number of cigarettes per day), the antecedent of conization and obstetric history;Adverse obstetric events in previous pregnancies: late miscarriage; threatened preterm labor (TPL), determined as cervical changes caused by regular uterine contractions occurring before 35 weeks and which required hospitalization and administration of tocolytic therapy; preterm premature rupture of membranes (PPROM); preterm delivery (PTB); or chorioamnionitis;Current pregnancy data: cervical length at the 20-week scan, urine culture results, *Group B Streptococcus* culture result, the episodes of symptomatic vaginal infections during pregnancy, the need for fetal lung maduration, and adverse obstetric events (described above);Delivery data: mode of delivery (and its reason in the case of cesarean section or instrumental delivery), gestational age at the time of delivery, intrapartum fever, postpartum endometritis;Newborn data: Apgar score, type of newborn resuscitation if needed, blood cord pH at birth, birth weight and neonatal morbidity or adverse neonatal events (admission in neonatal unit, admission to NICU, mechanical ventilation, respiratory distress, interventricular hemorrhage or retinopathy).

### 2.4. Statistical Analysis

Numerical variables were tested for normal distribution with the Kolmogorov–Smirov test. Medians and interquartile ranges (IQR) were used for describing numerical variables; frequencies and percentages for categorical ones. Prevalence estimates for vaginal infections were obtained in global (women with at least one of these infections) and in particular for each infection, with the corresponding 95% confidence intervals (95% CI).

The possible association of the type of vaginal flora of the patients (normal, intermediate or abnormal flora) with different outcomes (signs/symptoms, adverse obstetric and neonatal events) was analyzed using the Mann–Whitney’s U test or the Kruskal–Wallis test for numerical outcomes and the Pearson Chi-squared or the Fisher´s exact test for categorical outcomes, including all vaginal flora groups in the comparison as long as the sample size of each one ensured a reliable statistical analysis. A *p*-value below 0.05 was considered statistically significant; when this was the case, risk estimates (odds ratio, OR, with 95% CI) were obtained through logistic regression modelling.

Statistical tests were two-sided and were performed with SPSS V.20 (IBM Inc., Chica-go, IL, USA) and with Ime4 package in R, version 3.4 (RCoreTeam, 2017) [27].

## 3. Results

### 3.1. Recruitment

Figure 1 shows the recruitment process for the study. Finally, we obtained the complete birth data of 381 patients. 

Of the patients who wanted to participate and could not be included in the study, the majority were due to having recently been treated with antibiotics or progesterone, defined as an exclusion criterion. The main reason why some patients did not want to participate was the refusal of antibiotic treatment in case of pathological results. 

### 3.2. The Results of the Analyzed Smears of the Study Population (n = 400)

From the total of the samples analyzed, we identified 100 patients with abnormal flora (Nugent score between 7 and 10 or another pathogen detected in the cultures) and 9 patients with intermediate flora; which means that 27.2% of our patients presented alterations of the vaginal flora before 16 weeks of pregnancy.

Of all germs studied, the most frequently detected infection was *Ureaplasma urealyticum*, detected in 12.3% of patients screened and 49% of abnormal results, either in isolation or associated with other alterations. Two or more simultaneous infections were detected in 21 patients; in all these cases, *Ureaplasma urealyticum* was present since it is a bacterium that characteristically displaces the rest of the usual vaginal microbiota, favoring the appearance of other microorganisms or alterations of the flora. 

Other infections detected in our study were, in order of frequency: Candida in 47 patients (11.8% of the total), bacterial vaginosis in 20 patients (5%), *Mycoplasma hominis* in 5 patients (1.2%), and *Trichomonas vaginalis* in 3 patients (0.8%) (Table 1).

### 3.3. Comparative Analysis

#### 3.3.1. Signs and/or Symptoms at the Time of Sample Collection

Patients with an abnormal flora had a much higher probability of presenting vaginal signs and/or symptoms, such as leucorrhea, bad odor, itching or erythema, than patients with normal flora (OR 5.05, 95% CI 2.25–11.32, *p* < 0.001) (Table 2). For our usual practice, we can expect an altered result of flora in 24% of asymptomatic patients (81/338) versus 61% in patients presenting any vulvo-vaginal sign or symptom (17/28).

#### 3.3.2. Adverse Events in Previous Pregnancies

Of the 400 patients included only 219 had previous pregnancies that had exceeded the first trimester (there were 181 women nulliparas or who had first trimester miscarriages). It was noted that in the group of multipara patients with abnormal flora, the frequency of adverse events in previous pregnancies (late miscarriage, TPL, PPROM, PTB or chorioamnionitis) was more than double that in the group of multipara patients with normal flora (22.2% vs. 10.8%; OR 2.37, 95% CI 1.05–5.33, *p* = 0.034). None of the patients with intermediate flora had experienced adverse events in previous pregnancies (Table 3).

#### 3.3.3. Adverse Events in Current Pregnancy

Patients who experienced at least one adverse obstetric event in the current pregnancy were 9.2% of patients with abnormal flora versus 6.7% of patients with normal flora (*p* = 0.419 comparing 2 groups), while none of the patients with intermediate flora experienced adverse events. When analyzing the incidence of each specific adverse obstetric event independently, a risk of up to 5 times higher of preterm rupture of membranes was observed in patients with abnormal flora compared to patients with normal flora (5.3% vs. 1.1%, OR 5.11, 95% CI 1.20–21.71, *p* = 0.028). No statistically significant differences were observed for the rest of the adverse obstetric events in the current pregnancy (Table 4). In order to ensure that the above results were not biased by the difference between symptomatic patients (infection) and asymptomatic patients (colonization), we repeated the statistical analysis exclusively in asymptomatic patients, reaching very similar results and the same conclusions.

There were no statistically significant differences in birth weight or incidence of adverse neonatal events (Table 5) based on maternal vaginal flora.

## 4. Discussion

Our study provides information on the vaginal and endocervical flora alterations of women in their first trimester of pregnancy and the impact of early treatment on perinatal results. Previous studies in this field were focused on analyzing the association of specific infections (mainly due to BV and UU) with perinatal outcomes and, therefore, the usefulness of their targeted screening. However, in the present study, the most prevalent infections in our environment (BV, UU, *Mycoplasma hominis*, *Trichomonas vaginalis* and Candida) were screened in order to come closer to standard practice.

Among the findings, we must highlight that more than one of every four screened pregnant women had a vaginal or endocervical infection, and therefore, an increased risk of perinatal adverse events according to previous studies [5,6,7,8,9,10,11,12,13,14,15], and especially of preterm delivery [8,9,28]. This finding notes that vaginal flora alteration is a condition that affects our pregnant women considerably more frequently than other infections for which systematic screening are established, such as toxoplasmosis infection (affecting 0.1% of pregnancies) [29], cytomegalovirus infection (1%) [30] or asymptomatic bacteriuria (2–10%) [31]. 

When the perinatal outcomes in the current pregnancy were analyzed, no higher risk of late miscarriage or adverse neonatal events were observed among pregnant women with abnormal flora compared to pregnant women without vaginal flora alterations, nor for preterm delivery. The absence of differences between groups, especially for the latest outcome, may be due to the early screening and treatment of these patients, as Kiss et al. described in a prospective study where a significantly lower rate of preterm delivery was observed in a group of patients screened (by Gram stain) and treated (for BV, candidosis or TV) in the first trimester of pregnancy compared to a control group [7]. Haahr et al. reported similar results in a study about pregnant women with bacterial vaginosis in the first trimester, as the preterm delivery risk was significantly lower in the group treated with clindamycin compared to the group that received placebo [32]. Therefore, there is evidence of the benefits of early screening and treatment in reducing the extra risk of adverse events in patients with abnormal flora. 

The main study in opposition to systematic screening in low-risk patients was the PREMEVA study (early clindamycin for bacterial vaginosis), published in The Lancet medical journal in 2018 by Subtil et al. [16] The results showed no evidence of a reduction in the risk of late miscarriage and there were similar rates of spontaneous preterm delivery between treatment group and placebo group. However, this study was not free of controversy due to a delay in the publication of the definitive results, the possible poor diagnosis of BV [33] and a very low rate of preterm delivery (1%).

The latest systematic review was published in 2019 by Peelen et al. [20] Nine studies were eligible for inclusion in this review, six of them found an association between vaginal microbiota composition and preterm delivery and three studies did not find any association. Despite the contradictory results and the heterogeneity between studies, the authors are inclined to support the evidence of an association, given that this is the conclusion obtained in the most recent and best quality studies. The main conclusion they reach is that there is a lack of more molecular-based, culture-independent studies that analyze the relationship between the vaginal microbiota and PTB as an outcome.

Publications with inconsistent results have been attributed in meta-analyses and systematic reviews to differences in the antibiotic regimen, the timing of treatment (<22 weeks vs. ≥22 weeks of pregnancy), the performance of control smears and re-treatment of women with persistently positive cultures or the inclusion of women with risk factors for preterm birth not related to vaginal dysbiosis.

As the main result of our study, and even though all patients with vaginal flora alterations/infections received early treatment, the risk of preterm premature rupture of membranes in the group of patients with abnormal flora remained five times higher than in patients with normal flora. Further consideration should be given to the possible association of late miscarriages and abnormal flora, they were more frequently observed in the group with abnormal flora despite treatment but without being statisticaly significant probably due to a low frequency of this event in our patients.

The data of our study is consistent with the published literature; women with adverse events in previous pregnancies should be considered as high risk, and they will represent a subgroup of patients that would benefit from screening in the first trimester, since the risk factor of poor obstetric history will be added to the possible infection in the current pregnancy, worsening the prognosis. 

In addition in a priori low-risk patients with asymptomatic infection, our study shows favorable perinatal results (similar to patients with normal flora), which could be a consequence of the early treatment of these infections, so we are determined to continue investigating the convenience of implementing a systematic screening. New studies evaluating the potential risk of the treatments for the pregnant women and their fetuses will also be necessary. We must not underestimate the fact that although many vaginal infections are asymptomatic, some women may actually present symptoms but are not forthcoming unless they are asked explicitly. For this reason we consider that specific questions concerning vulvo-vaginal symptomatology should be included in our anamnesis and we ought to perform a more detailed vaginal examination of our patients, including smear sampling in case of abnormal discharge.

Regarding other lines of research in this field, there are multiple recent publications about the need for improved diagnostic tests for bacterial vaginosis (beyond Amsel criteria and Nugent score) and the development of point-of-care (POC) tests. The first part of the challenge is to determine which tests or which combination of criteria reaches a high enough sensitivity and specificity for the diagnosis of BV (molecular diagnostic methods; metabolomics and proteomics; immune system markers; computer algorithms, etc). The other challenge is the development of cost-effective diagnostic tests, preferably POC tests, since these methods are much cheaper considering that they do not require laboratory facilities or a qualified staff. Several POC diagnostic assays exist to diagnose BV, for instance, the OSOM^®^ BVBlue^®^ test is a chromogenic test that detects elevated levels of the sialidase enzyme (producted by Gardnerella vaginalis, Prevotella bivia, and Atopobium vaginae), and it is a test validated for use in asymptomatic pregnant women [34,35,36].

In many of these studies, the determination of vaginal pH (associated with biomarkers) improved the accuracy of the diagnosis. As an example, the State of Thuringia (Germany) in 2016, promoted a pH self-monitoring program in pregnant women which resulted in a decrease in the incidence of prematurity. This type of test was well received by women because it gave them some control over their pregnancies [37].

All this progress in research and technology is expected to completely change vaginal and endocervical screening programs in the future and hopefully to contribute to a reduction in preterm birth rates, a public health priority considering childhood morbidity and mortality related to this condition. In addition, if screening programs finally prove to be useful, there will be substantial cost savings. In 2006, Kiss et al. published a cost-effectiveness study in Vienna based on a screening and treatment program that (based on previous studies) would hypothetically decrease the preterm birth rate by 50%. The direct costs associated with each newborn under 1900 g and its follow up throughout the first 6 years of life amounted to an expected average of 60,262€. On the other hand, the total costs of the screening and treatment program were estimated to be 46€ per pregnancy. Overall, the total expected savings amounted to more than 11 million euros in one year, with screening and treatment costs representing only 7% of the direct costs saved as a result of the screening and treatment program [38].

To conclude, the main strength of our study lies in the prospective analysis carried out in a large sample of patients, secondly, the high patient adherence to follow up and treatment, and thirdly, the very little loss of data.

The principal limitations of our study were, firstly, not having performed control smears after the treatment of the infections to be sure that they were resolved, secondly, the small number of patients with intermediate flora (a group that has been omitted in many comparisons so as not to penalize the power of statistical analyses), and thirdly, the low frequency of some obstetric events that impaired determining their possible association with a particular vaginal flora profile. In addition, the absence of a control group diagnosed of an alteration of the flora or a vaginal infection and not treated impaired estimating the real impact of the treatment on the risk of adverse events. This untreated group would have contrary to ethical values since there is evidence that correlates these infections with poor obstetric outcomes. 

## 5. Conclusions

Given that prematurity is a major obstetric problem in developed countries and that the results of our study suggest that the early treatment of abnormal flora could improve prematurity rates, we should consider the benefits of implementing a screening policy in the first trimester. This would be especially important in patients with adverse events in previous pregnancies. We consider that specific questions concerning vulvo-vaginal symptomatology should be included in the anamnesis of the first visit of gestational follow up. Further studies, preferably with new diagnostic techniques, are needed to determine whether screening asymptomatic low-risk patients would be of benefit to perinatal outcomes.

## Figures and Tables

**Figure 1 pathogens-10-01610-f001:**
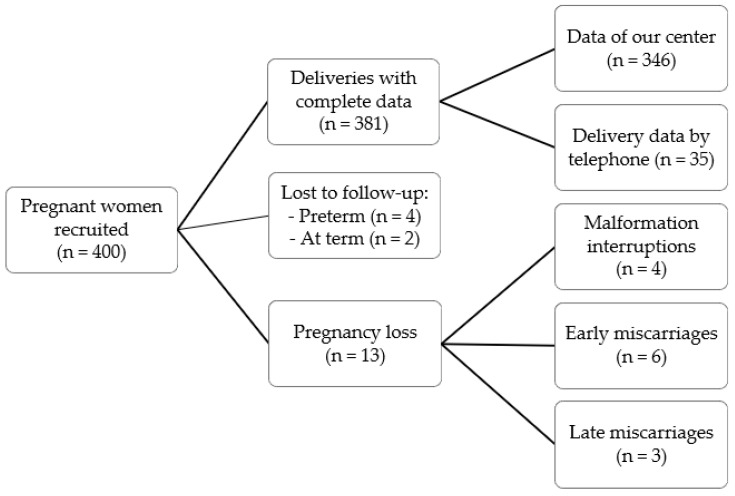
Flow chart of the study data.

**Table 1 pathogens-10-01610-t001:** Results of the smears of the study patients (*n* = 400).

Flora	Category		Number of Patients (%)	Total Percentage of Type of Flora
Normal			291 (72.8)	72.8
Intermediate			9 (2.2)	2.2
Abnormal	Ureaplasma urealyticum	Total: Isolated Associated with others ^2^	49 (12.3) 28 21	25.0
	*Candida* spp.	Total: Isolated Asociated with UU ^3^	47 (11.8) 37 10	
	Bacterial Vaginosis ^1^	Total: Isolated Asociated with UU ^3^	20 (5.0) 12 8	
	*Mycoplasma hominis*	Total: Isolated Asociated with UU ^3^	5 (1.2) 1 4	
	*Trichomonas vaginalis*	Total: Isolated Asociated with UU ^3^	3 (0.8) 1 2	

^1^ Nugent score between 7 and 10. ^2^ UU infections occurring in association with other infections. ^3^ In the remaining infections, those associated with UU.

**Table 2 pathogens-10-01610-t002:** Signs and/or symptoms of the patients at the time of sample collection in relation to the results of the smears (*n* = 366).

Signs and/or Symptoms ^a^	Normal Flora *n* = 268	Intermediate Flora *n* = 8	Abnormal Flora *n* = 90	*p*-Value 3 Groups	*p*-Value 2 Groups ^b^	OR (95%CI) 2 Groups ^b^
No Signs or Symptoms	257 (95.9)	7 (87.5)	74 (82.2)	<0.001 *	<0.001 *	5.05 (2.25–11.32)
Signs or Symptoms	11 (4.1)	1 (12.5)	16 (17.8)			

Data shown as *n* (% of total with data). ^a^ Leucorrhea, bad odor, itching or erythema. ^b^ Abnormal flora vs. normal flora. * Statistically significant differences.

**Table 3 pathogens-10-01610-t003:** Adverse events in previous pregnancies in relation to the results of the smears (*n* = 219).

Previous Obstetric Adverse Event ^a^	Normal Flora *n* = 158	Intermediate Flora *n* = 7	Abnormal Flora *n* = 54	*p*-Value 3 Groups	*p*-Value 2 Groups ^b^	OR (95% CI) 2 Groups ^b^
Yes	17 (10.8)	0 (0.0)	12 (22.2)	0.058	0.034 *	2.37 (1.05–5.33)
No	141 (89.2)	7 (100.0)	42 (77.8)			

Data shown as *n* (% of total with data). ^a^ Late miscarrriage, TPL, PPROM, preterm delivery, chorioamnionitis. ^b^ Abnormal flora vs. normal flora. * Statistically significant differences.

**Table 4 pathogens-10-01610-t004:** Adverse events in patients’ current pregnancy in relation to smear results.

Adverse Event	Normal Flora *n* = 283	Intermediate Flora *n* = 9	Abnormal Flora *n* = 98	*p*-Value 3 Groups	*p*-Value 2 Groups ^a^
Late Miscarriage	1 (0.4)	0/9 (0.0)	2 (2.0)	0.248	0.164
Chorioamnionitis	4 (1.4)	0/9 (0.0)	2 (2.0)	0.847	0.650
TPL	6/279 (2.2)	0/9 (0.0)	2/95 (2.1)	0.906	1.000
PPROM	3/279 (1.1)	0/9 (0.0)	5/95 (5.3)	0.043 *	0.028 *
Preterm Delivery	17/279 (6.1)	0/9 (0.0)	6/95 (6.3)	0.743	0.938

Data shown as *n* (% of total with data). * Statistically significant differences. ^a^ Abnormal vs. normal flora. The inclusion of a denominator indicates missing data for that particular characteristic. For the statistical calculations of late miscarriage and chorioamnionitis, we have the data of 390 patients (subtracting 6 early miscarriages and 4 malformation interruptions). For the statistical calculations of TPL, PPROM and preterm delivery, we have the data of 383 patients (subtracting 3 late miscarriages and 4 losses to follow up before reaching the term of pregnancy).

**Table 5 pathogens-10-01610-t005:** Relationship between birth weight, adverse neonatal events and smear results.

Birth Weight	Normal Flora *n* = 277	Intermediate Flora *n* = 9	Abnormal Flora *n* = 93	*p*-Value 3 Groups	*p*-Value 2 Groups ^a^
Weight in grams (median, IQR)	3266 (2980–3588)	3110 (3060–3578)	3288 (3056–3674)	0.318	0.144
Weight < 2500 g	15 (5.4)	0 (0.0)	2 (2.2)	0.339	0.259
Neonatal Morbidity					
Admission in neonatal unit	17 (6.1)	1 (11.1)	4 (4.2)	0.621	0.491
Admission to NICU	8 (2.9)	0 (0.0)	0 (0.0)	0.218	0.211
Mechanical ventilation	1 (0.4)	0 (0.0)	1 (1.1)	0.703	0.444
Respiratory distress	10 (3.6)	0 (0.0)	3 (3.2)	0.834	1.000
Interventricular hemorrhage	1 (0.4)	0 (0.0)	0 (0.0)	0.830	1.000
Retinopathy	1 (0.4)	0 (0.0)	0 (0.0)	0.830	1.000

Data shown as *n* (% of total with data). ^a^ Abnormal vs. normal flora.

## Data Availability

The datasets generated during and/or analysed during the current study are available from the corresponding author on reasonable request.

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
