# Peer review of "Screening for Vaginal and Endocervical Infections in the First Trimester of Pregnancy? A Study That Ignites an Old Debate"

_pathogens, 2021, doi:10.3390/pathogens10121610_

Round 1

Reviewer 1 Report

Dear editor, 

Thank you for inviting me to review the article of Leonie Toboso Silgo and colleagues. Their work described a prospective single-center study involving 40 women during the frist trimester of their pregnancy, who were screened and treated for vaginal and endocervical infections using Gram stain smears and cultural methods. The authors found that women an abnormal pathogen found on the screening smears had a 5-fold increased risk of pPROM, however without having an effect on preterm delivery or neonatal morbidity. The authors concluded that early treatment may explain the favorable perinatal outcomes in those with an infection, voting for a screening program in early pregnancy.

The topic of this article is timely, although a number of studies have already been conducted in this field. Nevertheless, currently available systematic reviews and Cochrane database analyses (Brocklehurst et al.) included only one prospective study (Kiss et al BMJ 2004) which was conducted in a outpatient (non-hospital) setting. Therefore, it is generally worth investing to further process this article to peer-review. Moreover, preterm birth is still the major causative factor of neonatal morbidity and mortality and preterm birth rates are stable high. Considering the fact that vaginal dysbiosis is one of the major factors for preterm delivery (about 40%, literature is pretty clear), this again underlines the importance of this article.

The manuscript has a number of flaws and needs to be extensive revised to ensure proper quality for publication (major revision). Indeed I would recommend to give the authors the possibility and time to revise the paper as the topic is of high importance and public health relevance. Please find my comments below.

General comments (apply to the entire manuscript/all sections):

*The paper is written in a clear manor, English language is fine, some sentences appear to be a little complicated. I suggest native language editing if this has not already been done.

*Change candidiasis to candidosis, as the ending -iasis is meant to describe parasitic diseases (applies to the entire manuscript)

*Please decide if you refer to “abnormal flora” or “bacterial vaginosis”, but do not change in between the terms (e.g. Line 159; applies to the entire manuscript)

*Please clearly give whether the patients you screened were asymptomatic or symptomatic. You should not combine both groups as screening refers to an asymptomatic group. This point is crucial and applies to the entire manuscript (applies to the entire manuscript).

*Please clarify whether you give information on contamination, colonization or infection: e.g. Line 250 – you refer to UU but did the patient really have symptoms? If not, this is not an infection but colonization. Please be more specific. This point is crucial and also applies to the entire manuscript.

*There should only be two decimal places after the comma. Please round all numbers throughout the manuscript (applies to the entire manuscript)

*Terms like “participants” should changed to “women”, “abortion” to “miscarriage”, “gestation” to “pregnancy”, “abnormal flora” to “bv” etc., generally they should be used consequently in the same manor without changes.

*The finding that pPROM was more frequent among women with BV, but that PTD was not, is counterintuitive. I would ask to authors to recalculate for the particular PTD groups (23-28 wks, 28-34 wks, 34-37 wks, 37+ wks).

*I have not checked the numbers – numbers should be thoroughly checked between abstract/mainbody/tables/figures to avoid rejection. Percentages should always give 100%. Please ensure this.

*The discussion section should not give numbers and percentages. Please revise without repeating the results (e.g., lines 254-298).

*After having revised the manuscript please ensure that the abstract is still a short presentation of the findings in your paper (e.g. revise conclusion to ensure correspondence between abstract’s and mainbody’s conclusion).

Minor comments:

*Line 42: please give a reference for your statement on costs (e.g. Kiss et al. EJOGRB 2006)

*Line 55: please change abortion to miscarriage

*Line 56: omit the word “so called”

*Line 60: in the study you are citing it was most importantly recurrent candidosis (change asymptomatic to recurrent)

*Lines 63-65: Please cite Subtil et al. Lancet 2018 here as well

*Lines 79-80: Revise the sentence, it sounds confusing

*Lines 79-87: How about antiseptics? Were these excluded as well? 

*Line 93: please include end of recruitment as well

*Lines 98-99: how did you evaluate the presence of leucorrhoea, etc.? Please provide more information on the method

*Lines 100-105: who evaluated the gram stained smears? Lab personel? How many different persons? Please give more details on the evaluator.

*Lines 107-113: please give Nugent scores for normal and BV as well

*Lines 111-113: why cultures? In whom did you perform cultures and at which step? Which method and product did you use? This is a crucial point.

*Lines 115-113: which guidelines did you treatment follow? Please cite.

*Line 134: “were recorded”

*Lines 140-142: please change abortion to miscarriage

*Line 146: It is not the type but the mode of delivery, include “in case of cesarean …”

*Line 152: “with the Kolmogorov-Smirnov test.”

*Line 155: “as a whole” is a very inelegant term – please revise

*Line 160-161: again please include “the” before the name of the test

*Figure 1: If possible please extend your flowchart to screening-exclusion-recruitment in the very beginning

*Line 179ff/Table 1: please do not give two decimal places after the comma but always just give one.

*Line 181: So as you found UU here – did you performed cultures in all patients?

*Table 1: bacterial vaginosis is not a pathogen. Do you refer to G.vaginalis? Please refer to the Candida strain here (C. albicans?). Frequencies should refer to N (for number).

*Line 192: with “an” abnormal flora (or BV, see above)

*Line 193: omit the word “those”

*Table 2: I would change “absents” and “presents” to “symptoms” and “no symptoms”

*All tables: sign of percentage (%) should be given in brackets

*Lines 203f: adverse events need to be defined in the M&M section

*Line 244: the study has already been conducted, please write in past term. Is the simple prevalence rate really the main result of your study?

*Table 4: how was “threatened preterm labor” defined? Due to what? Please define in the M&M section. Please change “gestation” to “pregnancy” in the heading of the table.

*Line 237: how were “adverse neonatal outcomes” defined? Please define in the M&M section.

*Table 5: typo “mechanical”, “admission to NICU”, change “smears” to “smear” in the heading of the table.

*Line 244f: is this really the main finding? I would suggest that the main finding is the higher likelihood for pPROM in women with an abnormal flora, but not the prevalence.

*Line 254f: do not change between the terms “patients”, “participants”, “cases” etc. Try to be more precise and clear.

*Lines 262f: I would not use consequent numbering (1-4) but conventional paragraphs. You could e.g. use words like “firstly” etc. to start the paragraphs.

*In all these four paragraphs the discussion with the commonly available literature is missing. Please thoroughly revise.

*Line 294: the plural of “flora” is flora but not “floras”

*Lines 300f: please end the sentence after “size”.

*Line 303: change “ultrasound” to “sonographic”

*Line 304: do you have data on the cervical length of your patients? Could you include this in the results or is this what you are referring to “threatened preterm labor”. If yes, this term should be changed to “cervical insufficiency”. This point is crucial.

*Lines 318-319: As screening is important, please include recent validation studies that showed feasibility of point-of-care tests for bv and vvc (Foessleitner et al. J Clin Med 2021, Foessleitner et al. J Fungi 2021). Please also include a short sentence of self-screening methods by e.g. pH measurement gloves as they are used in Thurinigia, Germany (Hoyme and Hesse. Arch Gynecol Obstet, 2021). Generally, your discussion is currently more of a summary of your findings than a discussion with literature.

*Lines 324-327: The conclusion is not correct. The main finding is, from my point of view and apart from the high prevalence, the increased risk for pPROM. You did not analyze cost effectiveness it this paper. The conclusion of the mainbody should be equal to that of the abstract.

*However it would be beneficial to shortly refer to cost effectiveness and include 1-2 sentences in the discussion section (ergo “why screening makes sense from a public health perspective”) and e.g. cite Kiss et al. Eur J Obstet Gynecol Reprod Biol 2006.

Reviewer 2 Report

ABSTRACT

  • In the results section, consider defining "abnormal flora"

INTRODUCTION

  • For the stated rate of prematurity (11-12%) is this globally or regionally? What countries are being referred to?
  • Line 54-56 should be revised, as the odds ratio for late abortion does not indicate double the odds of prematurity as it is currently written
  • It is unclear what specific vaginal/endocervical infections are being referred to in the last paragraph (lines 66-71). Will all vaginal/endocervical infections be examined? Will bacterial and/or fungal infections be explored? Will this only examine the most common vaginal/endocervical infections?

MATERIALS AND METHODS

  • Was the number of times a participant may have been pregnant before accounted for or were only first-time mothers included in the sample?
  • Additional details of the inclusion  criteria should be included
  • It is not clear if Nugent scoring was used to classify all samples collected (lines 107-113). The authors use three levels of classification of flora: normal, intermediate (Nugent score 4-6), and abnormal. However, normal flora could be indicated by a Nugent score of 1-3, whereas abnormal flora could also be considered Nugent score 4-10 and the presence of BV is indicated by Nugent score 7-10. Abnormal, for the purposes of this study, was also classified by the presence of infection. Therefore, was more than one method used for the classification of flora, i.e the absence/presence of infection and Nugent scoring? The classification of flora requires additional consideration as it is a bit ambiguous.
  • Were  Chlamydia trachomatis infections examined and treated? (lines 115-123)

RESULTS

  • Table 1- the percentage of total screened requires further explanation

DISCUSSION

  • Since your main variable of interest is alterations of flora, a more clear definition of what this looks like is warranted. It is still unclear if flora is being examined in terms of Nugent scoring or the absence/presence of infection or both. Consider the following scenarios: a) normal (NS 1-3), intermediate (NS 4-6), BV (NS 7-10); b) normal (NS 1-3), abnormal (NS 4-10); c) normal (absence of vaginal/endocervical infection), abnormal (presence of vaginal/endocervical infection); d) normal (NS 1-3 and absence of vaginal/endocervical infection), abnormal (NS 4-10 and presence of vaginal/endocervical infection), intermediate (NS 1-3 and presence of vaginal/endocervical infection OR NS 4-10 and absence of vaginal/endocervical infection). 
  • The discussion lacks a comparison to the available literature. How does this study align/differ from published research? What new information is added?

Round 2

Reviewer 2 Report

Great job responding to the comments